# Regulatory Macrophages and Tolerogenic Dendritic Cells in Myeloid Regulatory Cell-Based Therapies

**DOI:** 10.3390/ijms22157970

**Published:** 2021-07-26

**Authors:** Maaike Suuring, Aurélie Moreau

**Affiliations:** Centre de Recherche en Transplantation et Immunologie—UMR1064, INSERM—ITUN, Nantes Université, CHU Nantes, 44000 Nantes, France; maaike.suuring@univ-nantes.fr

**Keywords:** cell therapy, regulatory myeloid cells, tolerogenic dendritic cells, regulatory macrophages, clinical trials, safety

## Abstract

Myeloid regulatory cell-based therapy has been shown to be a promising cell-based medicinal approach in organ transplantation and for the treatment of autoimmune diseases, such as type 1 diabetes, rheumatoid arthritis, Crohn’s disease and multiple sclerosis. Dendritic cells (DCs) are the most efficient antigen-presenting cells and can naturally acquire tolerogenic properties through a variety of differentiation signals and stimuli. Several subtypes of DCs have been generated using additional agents, including vitamin D3, rapamycin and dexamethasone, or immunosuppressive cytokines, such as interleukin-10 (IL-10) and transforming growth factor-beta (TGF-β). These cells have been extensively studied in animals and humans to develop clinical-grade tolerogenic (tol)DCs. Regulatory macrophages (Mregs) are another type of protective myeloid cell that provide a tolerogenic environment, and have mainly been studied within the context of research on organ transplantation. This review aims to thoroughly describe the ex vivo generation of tolDCs and Mregs, their mechanism of action, as well as their therapeutic application and assessment in human clinical trials.

## 1. Introduction

New immunological therapies have been developing rapidly over the last few years, and one approach has focused on the immunoregulatory function of myeloid regulatory cells. Medawar’s earliest discovery (1953) showed that transferring regulatory cells from tolerant to naive animals resulted in significant tolerance to subsequent organ transplant. This phenomenon led other scientists to investigate the immune system’s discrimination between ‘self’ and ‘nonself’, and demonstrated a biological rather than chemical immunological affect [1]. Medawar showed that isolated splenocytes, kidney cells or testicular cells from adult mice could be intravenously infused into immunologically immature mice in utero or perinatally. These cells became permanently engrafted, and the resulting chimeric adult mice were able to permanently accept skin grafts or other tissues from a similar donor mouse strain [2]. Whilst these initial results suggested a potential for therapeutic application for humans [3], more than 70 years of subsequent research has highlighted many obstacles, including the identification, isolation and characterization of tolerogenic immunological donor cells, and the characterization of the inflammatory, transcriptomic, epigenomic, and metabolic aspects of the recipient’s immunological status or disease [4,5].

Immunological diseases are characterized by a disturbance of immune homeostasis and the immunological responses triggered by specific antigens. Current immunological therapies include the use of immunomodulators and immunosuppressants, varying from monoclonal antibodies to small molecules. Due to their nonspecific function, these interventions generally reduce the global immune response in the human body, and do not induce life-long peripheral tolerance. Therefore, patients often require life-long administration of these drugs which are associated with risk factors such as increased infections, toxicity and cancer development [6].

Whilst antigens and T and B lymphocytes were the focus of early studies, dendritic cells (DCs) have more recently been shown to be pivotal regulators of both adaptive and innate immune responses. DCs are derived from hematopoietic stem cells and are found in lymphoid and nonlymphoid organs. They are defined as mature or immature depending on their ability to stimulate T cells. The primary function of immature DCs in the blood and peripheral tissues is to capture, process and present antigens. Upon antigen capture, DCs undergo a process of “maturation”, defined by enhanced antigen processing, induction and presentation of class I and class II major histocompatibility complex (MHC) and costimulatory molecules (CD80/86) on their cell surface, and cytokine production and secretion. DCs then migrate to primary and secondary lymphoid organs, where they present processed antigens to naive T cells to induce immunity or tolerance.

Macrophages, on the other hand, are considered as important immune effector cells. Whilst immunologists initially focused mainly on their role in immunity, their pivotal role in homeostasis, including in the clearance of erythrocytes, cell debris and cells undergoing apoptosis, was underestimated [7]. Macrophage receptors involved in homeostasis include scavenger receptors, phosphatidyl serine receptors, the thrombospondin receptor, integrins and complement receptors [8]. Phagocytosis-mediated receptors either fail to induce cytokine production or actively inhibit it via inhibitory signals [9]. Macrophages are functionally classified as M1 or M2, with M1 being classically activated, and M2 alternatively activated. M2 macrophages are associated with wound healing, host defense and immune regulation [10,11]. Both DCs and macrophages are part of the mononuclear phagocyte system, which is part of both innate and cell-mediated immunity.

Pioneering studies by Ralph Steinman demonstrated the importance of DCs and other regulatory myeloid cells in the induction of peripheral tolerance, in addition to their ‘normal’ function in immunity [12]. In 2001, Steinman and colleagues showed that DEC-205 (CD205), an adsorptive endocytosis receptor, was a key DC receptor for delivering antigens to DCs in vivo [12]. When DCs were charged with steady-state antigens via DEC-205, the T cells exposed to the DC-presented antigens in vivo either disappeared or became anergic to antigenic restimulation. This important study suggested that when presented with steady-state antigens, a primary function of DCs is to maintain peripheral tolerance to self-antigens. Whilst it had already been proposed that immature DCs could induce tolerance to transplantation antigens and contact allergens (in vitro/in vivo) [13,14], the DEC-205 approach showed that DCs could induce profound antigen-specific tolerance in vivo [15]. This led to the possibility of isolating, targeting and expanding specific immune cell populations ex vivo to generate cell-based medicinal products (CBMPs) for the treatment of a range of immune diseases [16,17].

A number of lymphoid cell-based cancer immunotherapy strategies have been tested to restrict tumor progression by targeting tumor-associated antigens [18,19]. Alternatively, approaches to improve antigen-presenting cell (APC) abilities have been employed to re-educate the host immune system to recognize tumors [20,21,22]. The injection of unmodified DCs or DCs loaded with specific antigens has also been tested as a CBMP in several immunological diseases [21]. Type 1 diabetes (T1D) was the first disease that was tested using DCs presenting a specific antigen to reduce the effects of hyperglycemia. Hyperglycemia is a hallmark of diabetes and leads to dysregulated APC function, facilitates the differentiation of proinflammatory Th1/Th17 cells, suppresses Tregs and contributes to the breakdown of peripheral tolerance. Preventing or reversing type 1 diabetes using APCs presenting a specific antigen is a method which may be able to restore durable tolerance and reverse the immune-mediated loss of pancreatic β-cells. Here, the DCs promote immune tolerance by inducing hyporesponsiveness or selective apoptosis of autoreactive T cells, or by inducing Tregs and regulatory B cells [23]. Subsequently, these induced Tregs can convey immunoregulatory properties to other T cells and proinflammatory DCs, resulting in long-term maintenance of Ag-specific tolerance, also called infectious tolerance [23,24,25].

Knowledge of tolerogenic DCs (tolDCs) and regulatory macrophages (Mreg) has also been expanding over recent years, and has shown promising results in animal models. This has extended to several phase I clinical trials to assess tolDCs in the treatment of autoimmune diseases and the prevention of transplantation rejection [26,27,28,29,30,31,32,33]. Cell-based therapies are an attractive alternative to general immunosuppressive treatments due to the possibility of inducing antigen-specific immunological tolerance and the long-lived effect of self-reinforcing peripheral regulation.

This review is focused on the therapeutic application of tolDCs and Mregs. The different subsets of tolDC (VitD3-tolDC, Rapa-DC, DC10, Dex-tolDC and ATDC) and Mregs are described in terms of their generation, phenotype and function. In the last sections, we highlight the completed and ongoing clinical trials using tolDCs and Mregs to treat autoimmune diseases such as rheumatoid arthritis (RA), type 1 diabetes, multiple sclerosis, and Crohn’s disease, as well as to prevent transplantation rejection.

## 2. Tolerogenic Dendritic Cells Regulate Peripheral Tolerance

DCs are central regulators of innate and adaptive immune responses. In human blood, several DC subtypes have been identified: conventional (c)DCs (cDC1 and cDC2), plasmacytoid (p)DCs and inflammatory (i)DCs [4,25]. DCs recognize pathogen-associated molecular patterns (PAMPs) and damage-associated molecular patterns (DAMPs) via membrane-conjunction pattern recognition receptors (PRRs) such as Toll-like receptors (TLRs) and c-type lectins [4]. Activated cDCs secrete IL-12 and tumor necrosis factor (TNF)-α, and prime naive CD4^+^ and CD8^+^ T cells. In steady-state conditions, circulatory cDCs have an immature phenotype and maintain immunological homeostasis [34]. DCs are potent antigen-presenting cells with two possible fates: they can either be immunogenic or tolerogenic. The mechanisms underlying this molecular divergence remains only partially understood.

As outlined briefly above, DCs survey tissues and acquire self- and non- self protein antigens. They then migrate via the blood directly to the spleen and lymph nodes (LN) and ‘present’ antigens to T cells in the LN, to either stimulate T cells or induce Tregs [35]. DCs display these antigens as peptides via the major histocompatibility complex (MHC) proteins on their surfaces, and T cells recognize these complexes using their T cell receptors (TCRs). During full T-cell stimulation, DC costimulatory molecules (CD80/83/86) bind to T cell costimulatory receptors (CD28), activating PI3K or other pathways that eventually lead to increased nuclear levels of NF-κB and AP-1 (transcription factors) and cellular proliferation. DCs provide a third signal, and these cytokines determine the differentiation toward different functional subsets of T cells. Different T-helper cell populations are also recognized based on their function and cytokine production, with Th1 protecting against intracellular infections and Th17 being associated with various autoimmune diseases.

Tolerogenic DCs are known to induce tolerance via several contact-dependent and contact-independent mechanisms. TolDCs are characterized by low expression of costimulatory molecules (CD80, CD86 and CD83), upregulation of inhibitory and modulatory receptors, secretion of low pro-inflammatory and high levels of anti-inflammatory cytokines (Figure 1). The potency of tolDCs is linked to the inhibition of effector T cell responses and the induction of regulatory T cells (Tregs) [35,36]. TolDCs present antigens through normal T cell TCR and DC MHC interactions, but without costimulatory binding of T cell CD28 to DC CD80/86. This results in low production of IL-2 and no proliferation of the T cells. These T cells undergo clonal anergy and remain functionally in a hyporesponsive state. Another mechanism by which tolDCs induce tolerance is via T cell apoptosis. TolDCs express programmed death-ligand 1 (PD-L1) and PDL-2, an inhibitory surface receptor that binds to the T cell surface marker programmed cell death protein 1 (PD-1). This triggers the intracellular recruitment of SHP-1 and SHP-2 phosphatases to the TCR, and CD28 signaling pathways to promote clonal anergy and Treg differentiation [35,36]. Other contact-dependent mechanisms involved in maintaining tolerance via DCs are mediated by molecules such as Fas-L, or immunoglobulin-like transcripts (ILTs) and/or contact-independent mechanisms mediated by cytokines such as interleukin-10 (IL-10) and transforming growth factor-beta (TGF-β), or immunomodulatory molecules such as indoleamine-2,3-dioxygenase (IDO) and inducible nitric oxide synthase (iNOS) [17]. Some studies have suggested that a subtype of DCs induce tolerance, whereas other studies indicate that all DCs, depending on their maturation or activation, can drive tolerance [37,38]. Another important consideration is the stability of tolDCs, since DCs are essential for both tolerance and immunity. It is probable that nonstimulated tolDCs change their phenotype when injected into an immune stimulatory environment. Stimulating tolDCs into a semimature state via lipopolysaccharide (LPS) can potentially stabilize the phenotype and improve antigen presentation and migration [39]. A full understanding of the mechanisms that drive tolerogenic and immunogenic DC differentiation is essential for their application in new therapies. Nevertheless, tolDCs have long been recognized as a potential cell-based immunotherapy due to their efficient promotion of specific antigen tolerance and their ability to control immune responses and promote peripheral tolerance [36,40].

Human tolDCs have been generated (ex vivo) by exposing blood monocytes to growth factors, cytokines, genetic modification and pharmacological agents [41,42,43,44,45,46,47,48,49,50,51,52]. There are now several strategies for preparing clinical-grade, monocyte-derived tolDCs in patients suffering from autoimmune or inflammatory diseases [53]. Clinical grade human tolDCs are differentiated from peripheral blood monocytes using granulocytes-macrophage colony-stimulating factor (GM-CSF) and IL-4. This differentiation can be enhanced by additional ex vivo coculturing in the presence of pharmacological and biological agents [51,53]. Pharmacological agents used in this process include vitamin-D3, corticosteroids, rapamycin, cyclosporine, tacrolimus, aspirin, atorvastatin, retinoic acid, mycophenolic acid and minocycline [41,42,43,44,45,46,47,48]. Vitamin D3, rapamycin and dexamethasone have been extensively studied in animal and human studies to develop clinical-grade tolDCs. Immunosuppressive cytokines such as IL-10 and TGF-β have also been shown to promote tolerogenic properties in human tolDCs [49,50,51,52,53], and IL-10-induced tolDCs have been extensively studied in animal and human studies. Other cytokines known to induce tolDC differentiation include TNF-α, interferon (IFN)-γ, hepatocyte growth factor and IL-21 [54,55,56,57].

### 2.1. Vitamin D3-Induced Tolerogenic Dendritic Cells (VITD3-tolDC)

One approach to generate tolDCs is with the fat-soluble hormone vitamin D3 (VitD3), acquired from food or synthesized in the skin upon radiation with ultraviolet-B. In addition to its traditional role in calcium/phosphate and bone homeostasis, 1α,25-dihydroxy vitamin D3 (1α,25(OH)2D3), which is the active metabolite of vitamin D3, has been identified as a potent natural modulator of innate and adaptive immunity [58]. DCs treated with 1α,25(OH)2D3 do not differentiate or mature, locking the cells in a tolerogenic/immature state [59]. VitD3-treated DCs display tolerogenic properties, production of IL-10, low expression of class II MHC-mediated antigen presentation (e.g., HLA-DR) and low expression of costimulatory molecules (e.g., CD80, CD86) [60,61,62,63,64]. Vitamin D3-induced tolDCs (vitD3-tolDCs) are thought to develop their regulatory properties through a semimature profile, inhibit or reduce T cell responses, and switch the immune response to a TH2 profile [59,61,65,66,67]. VitD3-DCs also upregulate inhibitory molecules, such as programmed death-ligand (PD-L) 1, PD-L2 and immunoglobulin-like transcript (ILT)-3/-4, induce the production of regulatory cytokines including IL-10, CCL22, TGF-β and TNF-α, and decrease production of IL-12 [65,68]. Furthermore, vitD3-tolDCs are characterized by reduced NF-κB-mediated activity and increased mTOR-mediated glucose metabolism [61,69]. As a result, VitD3-tolDCs modify the behavior of T cells by inducing hyporesponsiveness and shifting T cell polarization from T helper (Th)1- and Th17-mediated inflammatory responses to Treg responses. In addition, vitamin D3 has a profound impact on the migratory properties of DCs to inflamed tissues through the upregulation of chemokine receptor CXCR3 [58]. However, other studies have suggested that LPS activation is required to stimulate the migratory activity and antigen presentation of tolDCs while maintaining their tolerant characterization. CCR7 direct DCs to migrate to T cell areas in secondary lymph nodes in respons to CCL19. LPS-activated tolDCs displayed a CCR7-dependent migration in response to CCL19, whereas nonactivated tolDCs expressed a low level of CCR7 and did not migrate [70]. The different outcomes in studies focused on DC migration can be explained by the different methods of DC, treatment and migration analysis.

More recently, it has been shown that VitD3-tolDCs generated from the monocytes of healthy volunteers or patients with relapsing-remitting multiple sclerosis (MS) have similar properties, including a semimature phenotype, an anti-inflammatory profile and a low capacity to induce allogeneic T cell proliferation. Furthermore, these cells seem to show potential for clinical application, since hyporesponsiveness of myelin-reactive T cells from patients with relapsing-remitting MS was observed when these T cells were cultured with autologous VitD3-tolDCs loaded with myelin peptides [71].

### 2.2. Dexamethasone/VitaminD3-Tolerogenic Dendritic Cells (DEX/VITD3-tolDC)

Glucocorticoids (GCs) are extensively used as immunosuppressive and anti-inflammatory agents in different clinical settings, with cortisone being the first GC administered to RA patients in 1948 [72]. GCs are relevant for the treatment of asthma, dermatitis, autoimmune diseases and inflammation. The immunosuppressive effects of GCs on DCs are primarily evidenced as interference in key inflammatory signaling pathways and transcriptional regulators, such as NK-κB and AP-1, leading to increased secretion of regulatory cytokines, inhibition of inflammatory cytokines and cellular immunity and induction of tolDCs [73]. The reprogramming of these cells into tolDCs drives T cell hyporesponsiveness and expansion of Tregs [74,75]. Several studies have investigated the generation of VitD3-tolDCs in combination with the glucocorticoid dexamethasone (Dex) to increase their tolerogenic potential [67]. TolDCs generated with Dex and other stimuli, namely IL-10, TGF-β, VitD3 or prostaglandin E2, have a stable phenotype, express high levels of MHC class II, CD80 and CD86, low expression of CD40, and express high levels of IL-10 and TGF-β, resulting in poor T cell stimulation and a switch to a more tolerogenic phenotype. More importantly, Dex-tolDCs express CD209 and have a low expression of CD1a and CD14. This has been associated with induction of Treg cells in patients with Crohn’s disease (CD) [30], suggesting that Dex-tolDCs have potential for therapeutic application in these patients.

Dex/VitD3-tolDCs have also been investigated in rheumatoid arthritis (RA), where the generation of tolDCs was compared between healthy volunteers and RA patients. A comparison of these Dex/VitD3-tolDCs showed a similar phenotype and function between the two groups [76]. To favor their migration to the draining lymph nodes and their antigen presentation to T cells, VitD3-DCs or Dex/VitD3-DCs can be matured in vitro with LPS. These cells are described as semimature DCs, and induce memory T cell hyporesponsiveness and naive T cell proliferation associated with low IFN-γ and high IL-10 production [70,77].

### 2.3. Rapamycin-Treated Dendritic Cells (RAPA-DC)

Rapamycin (RAPA) inhibits the mechanistic target of rapamycin (mTOR), a crucial immune system regulator. DCs generated with RAPA enrich CD4 forkhead box p3 (FoxP3^+^) regulatory T cells and induce T cell apoptosis by an unknown mechanism [78]. In contrast to VitD3-DCs, rapamycin-treated DCs (Rapa-DC) express CD83 and CD86 markers and produce low amounts of IL-10 and high levels of IL-12p40/p70, which are characteristics of a mature DC phenotype [79]. However, Rapa-DCs induce low-level proliferation of allogeneic T cells, similar to the other tolDC types [80]. Furthermore, Rapa-DCs secrete high levels of IL-12 after LPS stimulation, thereby promoting the induction of Treg Foxp3^+^ cells in mice. Rapa-DCs also promote experimental allograft survival, yet secrete high levels of IL-12, crucial for the generation of IFN-γ-CD4 T cells, thereby promoting the induction of Treg FoxP3^+^ cells in mice [78,79]. However, IFN-γ is pro-apoptotic, and IL-12-driven IFN-γ inhibits experimental graft-versus-host diseases (GVHD) [78].

Current knowledge suggests that high IL-12 secretion by DCs results in potent IFN-γ production by CD4^+^ T cells [81]. It has been suggested that increased production of IL-12 may be the underlying cause of reported mTOR inhibition and the occurrence of inflammatory disorders, including interstitial pneumonitis [82] and glomerulonephritis [83] in a subset of RAPA-treated transplant patients. However, post-transplantation of Rapa-DCs fails to enhance donor-specific T cell responses, even during infection. These findings emphasize the differential effects of rapamycin on T cell responses to pathogens and donor tissue, as well as the fact that many aspects of the mTOR signaling pathway in immune cells remain unknown [84]. Importantly, the increased IL-12 production by Rapa-DCs induces IFN-γ-mediated apoptosis in vitro and supports Treg induction. More precisely, it was shown that Rapa-DC-mediated apoptosis of alloreactive CD4^+^ T cells is IFN-γ-dependent, and results from the high production of IFN-γ by T cells that interact with Rapa-DCs [78].

### 2.4. Interleukin 10-Dendritic Cells (DC10)

IL-10 is another important and powerful anti-inflammatory cytokine that plays a pivotal role in diminishing the immune response and preventing chronic inflammatory pathologies [85]. Altered expression or deficiency of IL-10 enhances immune responses and leads to inflammatory bowel diseases and several autoimmune diseases [86,87]. Since its discovery, IL-10 has been shown to be produced by almost all leukocytes. IL-10 regulates the expression of several genes resulting in the downregulation of pro-inflammatory mediators, the inhibition of antigen (Ag) presentation and the upregulation of immune-modulatory molecules. Overall, IL-10 regulates APCs, inhibits effector T cell proliferation and cytokine production and promotes regulatory cell differentiation [88].

IL-10 can be used to generate two separate types of tolDCs depending on the cytokine exposure protocol. DCs exposed to IL-10 only at the end of the culture result in an immature tolDC phenotype and display resistance to maturation stimuli [89,90]. These DCs induce a state of anergy in CD4^+^ T cells and CD8^+^ T cells in an antigen-specific manner [90,91,92]. More recently, tolDCs derived from macaque monocytes in the presence of VitD3 and IL-10 were described as having tolerogenic properties, including resistance to maturation and low-level induction of T cell proliferation [93]. TolDCs generated from monocytes cocultured with IL-10 from the initiation of the culture, also called DC10, express CD83, CD80 and CD86, similar to mature DCs. In addition, DC10 also expresses Ig-like transcript (ILT)2, ILT3, ILT4 and HLA-G, secrete high levels of IL-10 and induce hyporesponsiveness of allogeneic T cells [52]. A key characteristic of DCs generated using IL-10 is their ability to influence the differentiation of Tr1 regulatory T cells [52,94]. Unfortunately, another property of IL-10-induced DCs is a decrease in trafficking of these cells to the lymph nodes. The chemokine CCR7 participates in the migration of DCs to the lymph nodes, and generating mouse DCs with IL-10 downregulates their expression of CCR7 and impairs their homing to lymph nodes in vivo [95]. In a mouse model of cardiac allotransplantation, it was shown that injection of DCs coexpressing IL-10 and CCR7 induced a significant prolongation of graft survival. However, DCs expressing only IL-10 or CCR7 did not affect heart allograft survival [96].

### 2.5. Autologous Tolerogenic Dendritic Cells (ATDC)

The conventional cytokines used to derive DCs from monocyte precursors are GM-CSF (granulocyte-macrophage-colony stimulating factor) and IL-4. However, in 2000, it was shown that tolDCs can be generated via GM-CSF stimulation only [97]. A study performed in mice showed that DCs generated with a low dose of GM-CSF in the absence of IL-4 had the properties of immature tolDCs. These cells had a high capacity to capture and present antigen, and induced low proliferation of allogeneic T cells. Furthermore, they resisted maturation and increased graft survival after in vivo injection [97]. Based on these studies, we focused on this simple protocol to derive tolerogenic bone marrow-derived rat DCs based on cell adherence and GM-CSF + IL-4 treatment (known as aBMDCs). Intravenous administration of these cells to rodents one day before transplantation showed an impressive prolongation of cardiac, skin or islet allograft survival, and was associated with the induction of regulatory T cells [97,98,99,100]. We next performed experiments to compare cells of different donor origins. Surprisingly, we observed that the injection of autologous aBMDCs induced a longer prolongation of heart allograft survival than injection of donor aBMDCs. Furthermore, rats receiving autologous aBMDCs and an allograft accepted a second donor skin graft, but not a third-party graft, demonstrating the development of robust regulatory mechanisms capable of maintaining donor-specific tolerance [101]. Additional studies conducted in mice and nonhuman primates have demonstrated the robustness of the low-dose GM-CSF protocol to generate autologous TolDC (ATDCs) [99,102]. The safety of ATDCs was tested in nonhuman primates, and the study showed that the intravenous administration was safe and well-tolerated. This successful in vivo proof of concept study has led to the generation of human ATDCs to evaluate their safety and efficacy in a phase I/II clinical trial [103].

Chitta et al. demonstrated that human monocyte-derived DCs generated in the presence of GM-CSF alone display similar tolerogenic properties to their mouse counterparts [104]. Similarly, we observed that human ATDCs display an immature phenotype by expressing low levels of costimulatory factors (CD80/86), low HLA-DR and low CD40. Furthermore, these ATDCs maintain their immature state after exposure to LPS or other TLR ligands. We also showed that these ATDCs expressed enhanced CCR7 after TLR stimulation, which lead to enhanced migration of these cells to secondary lymphoid organs, and importantly, maintenance of their tolerogenic potential. Furthermore, these ATDCs showed poor stimulation of allogeneic T cells and suppressed CD4^+^ T cell proliferation coculture with mature DCs. This suppression of T cell proliferation is associated with an inhibition of IFN-γ and IL-17 producing T cells. Furthermore, ATDCs were able to promote the expansion of Foxp3^+^ regulatory T cells. Unexpectedly, we demonstrated that ATDCs exhibit a high glycolytic rate resulting in a lactate-rich environment. Lactate produced by ATDCs was taken up by the CD4^+^ T cells dysregulating their aerobic glycolysis and suppressing their activation and proliferation [35]. Lastly, the therapeutic activity of human ATDCs was shown in a human-into-mouse graft-versus-host disease (GVHD) model performed in humanized mice. In this model, the increased survival rate observed in ATDC treated animals was associated with increased circulating lactate and a decrease in the proliferative capacity of splenic human CD4^+^ T cells.

To test these findings in a clinical trial, it was first necessary to develop a good manufacturing practice (GMP) compliant ATDCs production protocol. ATDCs generated from end-stage renal disease (ESRD) patients displayed similar profiles to healthy control ATDCs with no difference in the percentage of yield, expression of the main cell surface markers, maturation resistance after LPS exposure and the hypo stimulative and suppressive capacities of the ATDCs on T cells [36].

## 3. Regulatory Macrophages Regulating Peripheral Tolerance

Macrophages are first observed and expand in the yolk sac during primitive hematopoiesis. They are the only immune cell produced in the yolk sac, and are present as resident macrophages in some tissues. Subsequently, hematopoietic stem cells emerge in the fetal liver and give rise to all immune lineages during early development [105]. Monocytes originate from myeloid progenitor cells in the bone marrow and circulate in the blood for several days before entering tissues and differentiating into macrophages [106]. Monocyte-derived macrophages are at the centre of the immune response and have key roles in clearing pathogens and cell debris, antigen presentation and initiating adaptive immune responses [107]. Macrophages acquire specialized functions according to the stimuli present in the environment, which is referred to as macrophage polarization. Whilst the process of monocyte to macrophage differentiation is irreversible, macrophage polarization seems to be reversible [108]. This polarization allows macrophages to adopt a particular phenotype and functional program in response to specific signals such as interleukins, interferons, colony-stimulatory factors and tumor necrosis factors [109,110]. Macrophages can also produce several signals that alter their own physiology. There are two main populations of macrophages in vitro: classically activated macrophages (M1) and alternatively activated macrophages (M2) [111]. In vitro activation of macrophages has allowed a better understanding of the developmental requirements of different macrophage subsets, however, in vivo studies are complicated due to multiple stimuli within in the tissue environment.

Various stimuli control the polarization of macrophages and their subsequent expression of cytokine receptors, cell activation markers and cell adhesion molecules. PAMPs, DAMPs and pro-inflammatory cytokines such as interferon-γ (IFN-γ) and TNF-α prime macrophages toward a classical activation (M1) state [112,113]. The principal stimuli and markers of M1 activation are interleukin (IL)-1β, TNF-α, IL-6, IL-12, IL-23, CXCL9, CXCL10 and CXCL11 [108]. M1 macrophages secrete high concentrations of pro-inflammatory cytokines as well as high levels of superoxide anions and oxygen/nitrogen radicals which improves their killing capacity [114,115]. M1 derived NO reduces T cell proliferation and TNF-α blocks phagocytosis-mediated conversion of in vitro and in vivo inflammatory macrophages to the reparative M2 like phenotype [116,117].

Alternatively activated macrophages (M2) promote tissue repair, angiogenesis and tumor progression, and are also involved in immunoregulation and allergic reactions. Th2 cytokines are not inhibitors of classical activation, but participate in the induction of M2 macrophages. M2 macrophages typically produce high IL-10 and low IL-12 levels, and can be classified into at least four distinct subtypes based on different stimuli: M2a, M2b, M2c and M2d subtypes [118,119]. M2a are induced by IL-4 and IL13, and express high levels of the mannose receptor, the decoy receptor IL1RII and the L-1 receptor antagonist. M2b are induced following exposure to immune complexes and toll-like receptor agonists or IL-1 receptor ligands. They produce pro-inflammatory cytokines such as TNF-α, IL-1β and IL-6 [110]. M2c are induced by IL-10 and glucocorticoids, and express pentraxin-3 and high levels of TGF-β and IL-10. They also display high levels of the Mer receptor kinase, which is crucial for their efferocytotic activity. Finally, M2d are induced by TLR agonists that stimulate the adenosine receptor signaling pathway, which results in the suppression of TNF-α, IL-1 and IFN-γ expression.

M2 macrophages are also involved in wound healing, and are developed during innate or adaptive immune responses via IL-4 stimulation [120]. Indeed, IL-4 stimulates arginase (Arg) activity, allowing wound healing macrophages to convert arginine to orthinine resulting in increased extracellular matrix production [121,122]. L-ornithine is a precursor of proline and polyamines that promote collagen synthesis and cell proliferation, respectively, key aspects of tissue regeneration [123]. Interestingly, in murine macrophages, iNOS and Arg can compete for their common substrate, the amino acid L-arginine, which is key component of the urea cycle. L-arginine metabolism by iNOS produces L-citrulline and nitric oxide, a critical mediator of immunological and physiological aspects of tissue repair. Macrophages treated with IL-4 and/or IL-13 fail to present antigens to T cells, produce minimal pro-inflammatory cytokines and are less efficient at producing reactive oxygen and nitrogen radicals, and therefore, at killing intracellular pathogens [122]. Another population of macrophages are the regulatory macrophages [124]. The mutual characteristics of in vivo and in vitro regulatory macrophages include high IL-10 and low IL-12 secretion in assocation with the generation of Arg [121]. Regulatory macrophages express high levels of costimulatory molecules (CD80 and CD86) and can present antigens to T cells [11]. Recent metabolic studies have provided new insights into the precisely controlled metabolic changes in the different subtypes and macrophage states [107]. The specific metabolic changes are linked to effector mechanisms, for example, enhanced glycolysis and succinate levels, driving an inflammatory phenotype and fatty acid oxidation, causing an anti-inflammatory response [125,126].

Interestingly, in the context of tolerance induction, one particular metabolite, itaconate, has emerged as a regulator of macrophage function. Itaconate is synthesized from cis-aconitate in the tricarboxylic acid cycle (TCA cycle) in macrophages activated with different factors and signals [127]. It is required for the activation of the anti-inflammatory transcription factor (Nrf2). Itaconate alkylates cysteine residues, enabling Nrf2 to increase the expression of downstream genes with anti-oxidant and anti-inflammatory capacities. Type I interferons boost the expression of Irg1 and itaconate production. However, itaconate limits the type I interferon response, resulting in a negative feedback loop [128]. Evidence also points to itaconate being an anti-inflammatory factor, acting similarly to the immunosuppressive cytokine IL-10 [107]. Peroxisome proliferator-activated receptor-γ (PPARγ), which regulates adipogenesis and fat cell function, is involved in M2 macrophage activation. PPARγ has been shown to be required for IL-4-dependent gene expression, and PPARγ deficiency leads to enhanced expression of IRG1 [129]. It is possible that when PPARγ is blocked, itaconate increases to restore homeostasis. Conversely, lowering itaconate may drive oxidative phosphorylation, which is a hallmark of M2 macrophages. The therapeutic applications of itaconate are most likely centered on its ability to regulate macrophage anti-inflammatory responses [107]. Another study showed that suppressive macrophages can also be generated in grafted mice cotreated to block costimulatory signaling. Indeed, a blocking anti-CD40L mAb resulted in accumulation of CD11b^+^, CD115^+^, DC-SIGN^+^ macrophages in the allograft, promoting the expansion of Tregs while inhibiting CD8^+^ T cell accumulation. These DC-SIGN macrophages produce immuno-regulatory IL-10, and their in vivo accumulation is controlled by M-CSF [117].

Similar to TolDCs, regulatory macrophages have been generated in vitro for use as CBMP for immunological diseases. The M1 polarization state is directed by stimulation using IFN-γ or LPS. However, the longstanding view that IFN-γ acts as a predominantly inflammatory cytokine is challenged by findings of its direct and indirect regulatory effects. IFN-γ can regulate the tolerogenic effects of both innate and adaptive immune cells, promoting tolerance of APC and inducing Treg differentiation [130]. A prominent role of IFN-γ is the induction of IDO expression on DC. The priming effect of IFN-γ on DC maturation by LPS is subjected to exhaustion; when DC pro-inflammatory cytokines decrease, IFN-γ induces IDO competence. Continuous IDO expression by DCs is necessary for an immunosuppressive state, which is initiated and maintained by an IFN-γ-dependent kynurenin-AhR pathway. A study from 10 years ago demonstrated a therapeutic effect of IFN-γ-stimulated DCs in an experimental model of allergic encephalomyeltitis [131]. These DCs were immature, showing low costimulatory molecule expression (CD80/86) and low MHC class II, with increased IDO expression [132]. IDO is an enzyme that catabolizes tryptophan into kynerenines. Expression of IDO by DCs reduces the amount of tryptophan in the microenvironmentand consequently inhibits T cell proliferation via stress-response pathways because T cells are sensitive to amino-acid withdrawal. Kynurenine metabolites also contribute to the induction of FoxP3^+^ Tregs and suppression of effector T cells [133]. Mosser et al. described a distinct active macrophage phenotype induced by IFN-γ, which resulted in macrophage activation with high IL-10 levels and low IL-12 levels. Analysis of gene expression allowed these activated macrophages to be differentiated from conventionally activated macrophages, or IL-4-stimulated macrophages, and these macrophages are classified as regulatory macrophages (Mregs) [9,134,135]. It was reported that mouse Mregs show a suppressive phenotype, and injection of these cells in transplanted mice extends allograft survival [136]. These Mregs are able to inhibit T cell proliferation and enhance graft survival and function through an iNOS dependent mechanism and by the release of anti-inflammatory factors [136,137,138]. These mouse Mregs displayed a unique state of differentiation distinguished by stable cell surface DHRS9 expression and potent suppressive functions [139].

Human Mregs are derived from CD14^+^ peripheral blood monocytes. They are stimulated with macrophage colony-stimulating factor (MCSF) for six days before stimulation with IFN-γ [140], and require contact with a plastic surface, ligation of FcγRIII (CD16) by serum immunoglobulin and exposure to other serum factors [140,141,142]. Human Mregs suppress mitogen-stimulated allogeneic T cell proliferation in vitro through interferon (IFN)-γ-induced IDO activity, as well as mediating a contact-dependent deletion of activated allogeneic T cells [137]. Because of their stable suppressive phenotype, human Mregs have potential as a CBMP for immunosuppressive, anti-inflammatory or tissue-reparative therapy. In mouse studies, intravenously injected allogeneic Mregs rapidly accumulated in the lungs, liver and spleen (but not LNs), and a proportion of these cells persisted for up to 4 weeks [139]. Importantly, these experiments showed that viable Mregs exerted a graft-protective effect that endured beyond their lifespan, which was mediated by recipient T cells [139]. Although mouse and human Mregs suppress allogeneic T cell proliferation and effector functions in vitro, the hypothesis that they can control recipient alloimmune responses in vivo through direct interaction of transferred cells with recipient T cells remains controversial. Human Mregs have been shown to increase Treg frequencies in peripheral blood in prospective kidney transplant recipients [139]. Taken together, these findings suggest that induced Tregs (iTreg) elicited by allogeneic Mreg therapy could play a mechanistically important role in its tolerogenic effects. Overall, Mreg therapy shows favorable results in human patients receiving a subsequent kidney graft from the same donor [139].

## 4. Completed and Ongoing Clinical Trials Investigating Tolerogenic Application of Myeloid Regulatory Cells in Autoimmune Diseases

Our increasing knowledge of tolDC biology and promising results from animal models have led to several phase I clinical trials to assess the safety and efficacy of tolDC therapies for autoimmunity. Clinical trials using immunogenic DCs have been developed over the last 28 years, whereas tolDC and Mreg therapies are recently emerging in the clinical area [143]. This was initiated by a pioneering study published in 2001, demonstrating the safety of injecting autologous immature DCs into healthy volunteers [144]. Injections of these DCs via the subcutaneous route were well-tolerated without signs of toxicity or development of autoimmunity. Inhibition of antigen-specific effector T cell function and induction of antigen-specific CD8^+^ Tregs in vivo was detected in the volunteers receiving the injection [144,145].

Clinical trials using tolerogenic immune cells to restore peripheral tolerance in patients with autoimmune diseases, such as RA, type1 diabetes, MS, and CD [26,27,28,29,30,31,32,33], have been completed or are ongoing (Figure 2). Overall, these trials showed that the therapy was well-tolerated, and no therapy-related reactions were observed (Table 1) [28,29,30,33].

The first phase I clinical trial using tolDCs included ten patients with type 1 diabetes. Three patients received control MoDCs, and seven patients received tolDCs generated by GMCSF/IL-4 stimulation and antisense oligonucleotides targeting CD40, CD86 and CD80 transcripts [146]. Testing of this approach in a preclinical NOD mouse model demonstrated prevention and reversal of type 1 diabetes, with the immunosuppressive DCs leading to proliferation and survival of Tregs [147]. In the clinical trial, the safety and tolerance of intradermal injections of both tolDCs and MoDCs was demonstrated. DCs upregulated the frequency of a potentially beneficial B220^+^CD11c-B cell population, at least in type 1 diabetes autoimmunity [26]. Following this, three other clinical trials that are either completed or ongoing have focused on type 1 diabetes, where they tested proinsulin-loaded VitD3-tolDCs, monocyte-derived DCs treated with antisense oligonucleotides targeting CD80, CD86, CD40 and autologous dendritic cell therapy (see Table 1). In a first-in-man dose-escalation phase 1 trial, nine patients with long-standing type 1 diabetes were intradermally administered proinsulin peptide-loaded VitD3-tolDCs. The preliminary results confirmed the feasibility and safety of this approach. However, testing patients with a shorter diagnosis of type 1 diabetes and preserved C-peptide production is warranted [148].

The treatment of RA using tolDCs has also been assessed in several phase I clinical trials. These have included testing the safety and efficacy of different tolDC subtypes, different preloaded antigens and various administration routes and schemes. Dex-tolDCs injected intra-arterially (single injection) were safe and well-tolerated, and early preliminary data suggested a potential beneficial effect [27]. A second study used Dex/VitD3 tolDCs loaded with autologous synovial fluid as a source of relevant autoantigens, enabling the treatment of both patients with seropositive RA and patients with seronegative RA, as well as other types of arthritis [29]. Participants had inflammatory arthritis of at least six months duration, including an inflamed knee joint with effusion and at least 30 min early morning stiffness. In addition, they had failed at least one disease-modifying antirheumatic drug (DMARD), including current therapy. TolDCs were added to stable DMARD and anti-inflammatory medicines. The Dex/VitD3 tolDCs loaded with autologous synovial fluid were administered in a single intralesional injection. Based on this unblinded phase I trial, tolDC therapy proved safe and worthy of further investigation. This conclusion is based on the absence of protocol-defined target knee flares and anecdotal evidence of improvement in the highest dose cohort participants. There were three knee flares recorded 7–10 days post-tolDC administration, but they occurred in the lower dose cohorts and are therefore more likely to reflect the natural history of knee synovitis following joint irrigation [29]. The third tolDC phase I clinical trial examined NF-κB inhibitor-treated tolDCs loaded with four citrullinated peptide antigens. Preclinical animal studies testing NF-κB inhibitor-treated DCs demonstrated that the delivery of immunomodulatory DCs exposed to autoantigens could suppress experimental arthritis in an antigen-specific manner [149]. However, translation to clinical trials is challenging because multiple autoantigens have been described in RA to which T cell responses are difficult to measure in vitro [149,150,151]. Disease-specific autoantibodies for citrullinated peptide antigens (ACPA) are found in the serum of 70% of RA patients and are strongly associated with HLA-DRB1 SE alleles [152]. Circulating autoreactive T cells recognizing citrullinated autoantigens were identified in the peripheral blood of HLA-DRB1 SE^+^ RA patients, suggesting that these patients are an appropriate target group [153,154,155]. Multiple genes driving NF-κB pathway activation are associated with RA susceptibility. Suppressing NF-κB in DCs can be achieved in vitro with Bay11-7082, a specific irreversible inhibitor of NF-κB, and these have been tested in mice and humans [156,157,158]. The NF-κB inhibitor-treated DCs were tested in 18 ACPA^+^ RA patients, 9 of whom were intradermally injected with low-dose and 9 with high-dose Rheumavax [28]. The peptides delivered by Rheumavax would be predicted to bind to the different HLA-DR SE molecules in the patients’ blood but may not attach to all HLA-DR molecules. Therefore, using anchor residue frequencies, citrullinated peptides could be further optimized and specifically modified for targeted binding to RA-associated HLA-DR alleles in individual patients. The preliminary results showed that a single intradermal injection of the NF-κB tolDCs was safe and that immunoregulatory and anti-inflammatory effects were observed in HLA risk genotype-positive RA patients.

Lastly, another phase 1 trial tested DCs pulsed with PAD4, NHRNPA2B1, citrullinated filaggrin and vimentin antigens in cohorts of three participants receiving three different doses of tolDC arthroscopically. No target knee flares were observed within five days of treatment. The IA tolDC therapy appears safe, feasible and acceptable, where the knee symptoms stabilized in two patients receiving the high doses of tolDC. However, no systemic clinical or immunomodulatory effects were detected [29]. In conclusion, the administration of the different subsets of tolDCs to RA patients was shown to be safe, well-tolerated and to have a potential beneficial effect. However, further studies are required to assess the clinical efficacy and associated immune effects of antigen-specific immunotherapy for RA [27,28,29].

The target antigen for the treatment of MS is not known. However, proteins within the myelin sheath, such as myelin basic protein (MBP), myelin oligodendrocyte glycoprotein (MOG) and myelin proteolipid (PLP), are important targets of an autoreactive immune response [159,160,161]. To develop antigen-specific tolerance for MS, several myelin immunodominant peptides have been identified ex vivo from patients with MS and tested in clinical trials using autologous DCs coupled with such peptides [160,162,163,164].

Treatment for the autoimmune disease MS has been tested in three phase I clinical trials, one completed, where two different subsets, Dex-tolDCs loaded with myelin peptides or aquaporin-4-derived peptides (AQP4) and VitD3-tolDCs loaded with several myelin peptides, were tested. The Dex-tolDCs loaded with myelin peptides or AQP4 were injected intravenously via three bi-weekly injections. This trial showed the feasibility and safety of treating patients with MS and neuromyelitis optica spectrum disorders (NMSOD) with the highest dose of viable tolDCs loaded with either myelin antigens or AQP4 antigens. The study design combined patients with MS and NMOSD, treating both groups for tolerance induction with seven myelin peptides while also adding AQP4 peptides for the NMOSD patient cohort. This design allowed leveraging treatments for both conditions. The Dex-tolDC clinical trial has been completed and shown that the therapy with CNS peptide-specific loaded tolDCs can regulate immune tolerance via the induction of regulatory T cell (Tr1) activity [30]. More recently, the potential of VitD3 to generate tolDCs from MS patients has been demonstrated. VitD3-tolDCs from MS patients displayed a maturation resistant phenotype even after rechallenge with an inflammatory stimulus. VitD3-tolDCs from MS patients are capable of inducing stable and antigen-specific T cell hyporesponsiveness. After in vitro stimulation with myelin-derived peptide-pulsed VitD3-tolDCs, the T cells were unresponsive to the myelin peptides used while retaining their capacity to respond to an unrelated antigen [71,80,164]. In the current clinical trial, harmonization of clinical, MRI and immunological evaluations of the patients will enable comparison of results between two phase I clinical trials evaluating the safety and feasibility of autologous tolDC administration in patients with active MS. The two VitD3-tolDC phase I/IIa clinical trials are ongoing but differ in the tolDC administration route, i.e., intranodally versus intradermally [31].

The need for new therapeutic tools for patients with refractory CD has also been assessed using immunotherapy strategies. Whilst the feasibility and efficacy of autologous stem cell transplantation has been shown, safety-related issues may limit its clinical application [165]. However, intestinal macrophages and DCs generate a specialized network of cells involved in sensing external pathogens while maintaining intestinal homeostasis. Intestinal DCs are crucially involved in supporting the inflammation in CD by producing pro-inflammatory cytokines IL-12, IL-23, and TNF-α in the absence of anti-inflammatory cytokines. IL-10 is critically involved in maintaining intestinal homeostasis. Importantly, in the absence of IL-10, mice spontaneously develop colitis, and in humans, mutations in the IL-10 pathway are associated with the development of severe early-onset CD [166]. In animal models, it has been observed that vasoactive intestinal peptide (VIP), an immunomodulatory neuropeptide released in inflammatory conditions, can generate tolDCs that can restore tolerance in vivo in trinitrobenzene sulphonic acid (TNBS)-induced colitis [167,168]. The generation of clinical-grade tolDCs from healthy and Crohn’s disease patients’ human monocytes identified that the combination of dexamethasone plus a cocktail of cytokines yielded stable tolDCs characterized by a semimature phenotype and high levels of IL-10 production [169]. Two-phase I clinical trials set out to test the safety and efficacy of Dex/VitA-tolDCs and Dex-tolDCs. The Dex/VitA-tolDC trial is completed, whereas the second trial using Dex-tolDCs was terminated due to low recruitment. The Dex/VitA-tolDCs were administered in a single intraperitoneal injection versus three injections administered bi-weekly. Nine CD patients were included in the trial and did not show any adverse effects due to the infusion. However, three patients withdrew from the study due to worsening CD, while the other patients showed improvement. One patient reached remission at 12 weeks, and for the other two cases, the quality of life improved. Intraperitoneal administration of autologous tolDCs appears safe and feasible in refractory CD patients. Future studies should be developed to test clinical benefit, determine the optimal administration route and dose, and monitor the immune responses [33].

## 5. Completed and Ongoing Clinical Trials Investigating Tolerogenic Application of Myeloid Regulatory Cells in Transplantation

Recent studies have shown tolDCs to be a promising method for enhancing transplantation survival while reducing patient dependency on immunosuppressive drugs [71,76,80,170,171]. Testing ex vivo generated tolDCs in human transplantation is a compelling application for several reasons. It induces Ag-specific T cell unresponsiveness promoting organ allograft survival and regulates memory T cell responses which represents a major barrier for long-term graft survival [144,145,172]. Furthermore, reports in the last 20 years have shown that in vitro generated immature allogeneic DCs, injected before transplantation, prolong heart and pancreatic islet allograft survival [88,146,147,148,149,150]. In the absence of immunosuppressive therapy, systemic administration of immature donor-derived DCs before transplantation prolonged allograft survival but never induced tolerance [88,147]. Another study described permanent heart allograft acceptance after injection of immature donor DCs, which were resistant to typical maturation-inducing stimuli seven days before transplantation [97]. Studies in mouse and primate models have shown that the best kinetics for allograft survival is a single DC injection one a week before transplantation [97,173]. However, in most clinical situations, the donor is known only a few hours before transplantation, and therefore these protocols would not be applicable in the clinical setting. A study from our group has shown that the effect of immature donor BMDCs on heart allograft survival was limited, but was more efficient than pregraft donor blood transfusion [98]. Importantly, we found that autologous immature BMDCs were more efficient than donor-derived ones. The generation of autologous tolDCs (ATDCs) from a patient’s own monocytes and injection peri-transplantation is clearly clinically feasible and potentially beneficial [98]. Following this observation, human ATDCs were generated as described previously (section ATDCs) and were transferred to a GMP facility [36]. In a completed phase I/IIa clinical trial [32], tolDCs (ATDCs) were intravenously administered once one day before kidney transplantation (Figure 2 and Table 2). Patients also received standard immunosuppression (tacrolimus, mycophenolate mofetil and prednisolone).

The administration of ATDCs was shown to be safe and efficient [32] In the same consortium as the ATDC clinical trials, called the ONE study, the safety of Mregs was also evaluated. The ONE study consisted of seven single-arm trials conducted internationally at eight different hospitals with a 60-week follow-up. These preclinical studies showed that preoperative administration of donor-derived Mregs via intravenous injection prolonged allograft survival in fully allogeneic recipients without lymphoid depletion, immunosuprresion or other conditioning regimens (Table 2) [141]. Importantly, this allograft-protective effect was not simply due to alloantigen exposure but depended upon Mregs expressing inducible nitric oxide synthase. One of the suggested mechanisms by which macrophages mediate graft loss is nitric oxide production contributing to endothelial cell cytotoxicity and renal tubular injury [174]. Mregs have been tested in a phase I clinical trial (ONEmreg12trial) as a means of safely minimizing maintenance immunosuppression in kidney transplant recipients [32].

Two other clinical trials in transplantation are ongoing (Table 2). Donor-derived VitD3-IL10-tolDCs are administered and studied for their safety following infusion into prospective living donor liver transplantation (LDLT) and kidney transplantation recipients (Figure 2). In this study, it was shown that prior to transplantation, donor-derived VitD3-IL10-tolDCs induced enhanced ‘expression’ of HLA and immunoregulatory molecules on circulating small extracellular vesicles (sEVs), together with cross-dressing to recipient APCs. TolDC infusion also led to alterations in CD8^+^ T memory and Treg populations post tolDC infusion and prior to graft implantation, with no evidence of pretransplant sensitization to donors [175]. In the future, this study will follow up on the patients who received VitD3-IL10-tolDCs before transplantation. They will report on the findings regarding the safety and efficacy of donor-derived VitD3-IL10-tolDCs in transplantation.

## 6. Future Perspectives of a Myeloid-Regulated Tolerogenic Environment in Immunological Therapies

Strategies for manufacturing clinical-grade myeloid regulatory cells have only emerged recently, and clinical application is today being widely investigated for autoimmune diseases and transplantation. Thus far, recent clinical trials of the different subsets of tolDC demonstrated the safety and feasibility of tolDC and Mreg-based cell therapies. Nevertheless, the stability of infused tolDC and Mreg products and the maintenance of their tolerogenic properties in vivo remain open issues to be tackled in order to improve the safety and the efficacy of these therapies. Over recent years, tolDCs and their function have been extensively studied and represent good candidates for DC-based treatments. They modulate effector immune responses, including effector T cells, while leading to long-term tolerance via in vivo induction of Tregs [88]. Knowledge of the specific pathways involved and the tolerogenic properties of these immune cells has been significantly enhanced by more recent transcriptomic and epigenomic characterizations.

Moreover, the number of tolDC subsets being described continues to increase and the optimal subset to be used as a medicinal product must be defined for each application in different pathologies and micro-environments. A comparative analysis of diverse populations of in vitro-differentiated tolDCs examining their stability, cytokine production profile, and suppressive activity indicated that all different subsets of myeloid regulatory cells have beneficial properties in maintaining a tolerogenic environment [88]. TolDC-based therapy can restore the immune system to a more homeostatic level, thereby decreasing the adverse side-effects of conventional immunosuppressive drugs [32]. The next goal will be to better understand the properties and application of tolDCs concerning immunological diseases. New methods of single-cell analysis can interpret the precise communication between immune cells [176]. Recent studies have also shown that immune metabolism and the micro-environment plays a pivotal role in controlling the functionality and properties of tolDCs. Epigenetic alterations for future therapies may provide novel targets to modulate the state of tolDCs [4]. Therefore, immune cell therapy in immunological diseases demands further investigation in the application of myeloid regulatory cells and their effect in vivo via larger clinical trials [32].

## Figures and Tables

**Figure 1 ijms-22-07970-f001:**
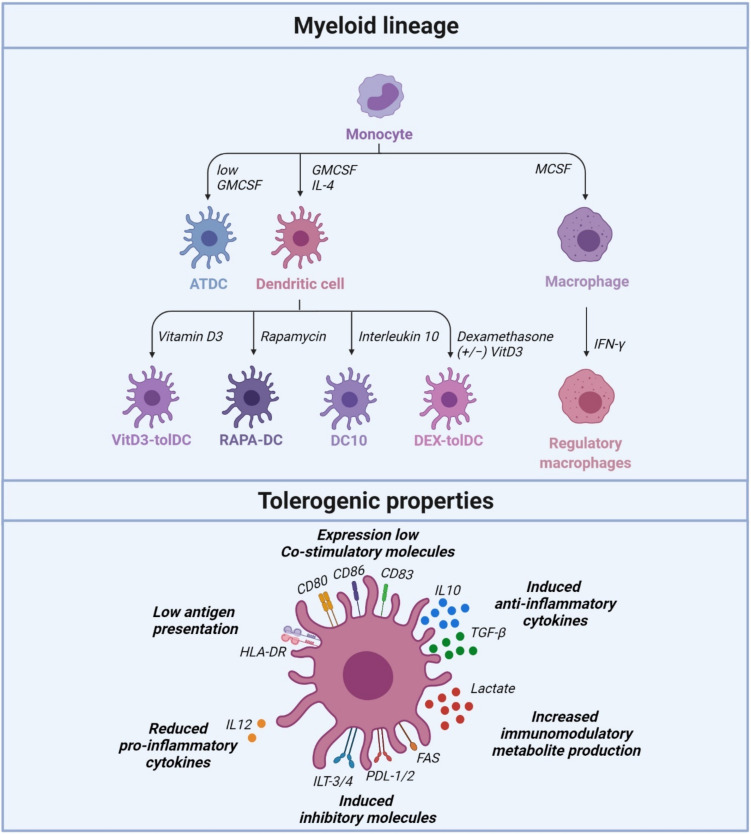
Overview of human tolDC generation by exposing blood monocytes to standard stimulation of granulocyte-macrophage colony-stimulating factor (GMCSF) and interleukin 4 (IL-4), plus additional growth factors, cytokines and pharmacological agents. Several strategies have been used to prepare clinical-grade, monocyte-derived tolDCs, where the stimuli, i.e., vitamin D3, rapamycin, dexamethasone and interleukin-10, have been extensively studied. These stimulated tolDCs, i.e., vitamin D3-tolDC (VitD3-tolDC), rapamycin-DC (RAPA-DC), interleukin-10 DC (DC10), dexamethasone tolDC (Dex-tolDC) and autologous tolDC (ATDC), have overlapping tolerogenic properties. TolDCs have low expression of costimulatory molecules (CD80, CD86 and CD83), induce immunosuppressive cytokines (like interleukin (IL)-10 and transforming growth factor (TGF)-β), induce inhibitory molecules, express a low antigen presentation molecules (Human leukocyte antigen (HLA)-DR) and lower secretion of interleukin (IL)-12. Created with BioRender.com.

**Figure 2 ijms-22-07970-f002:**
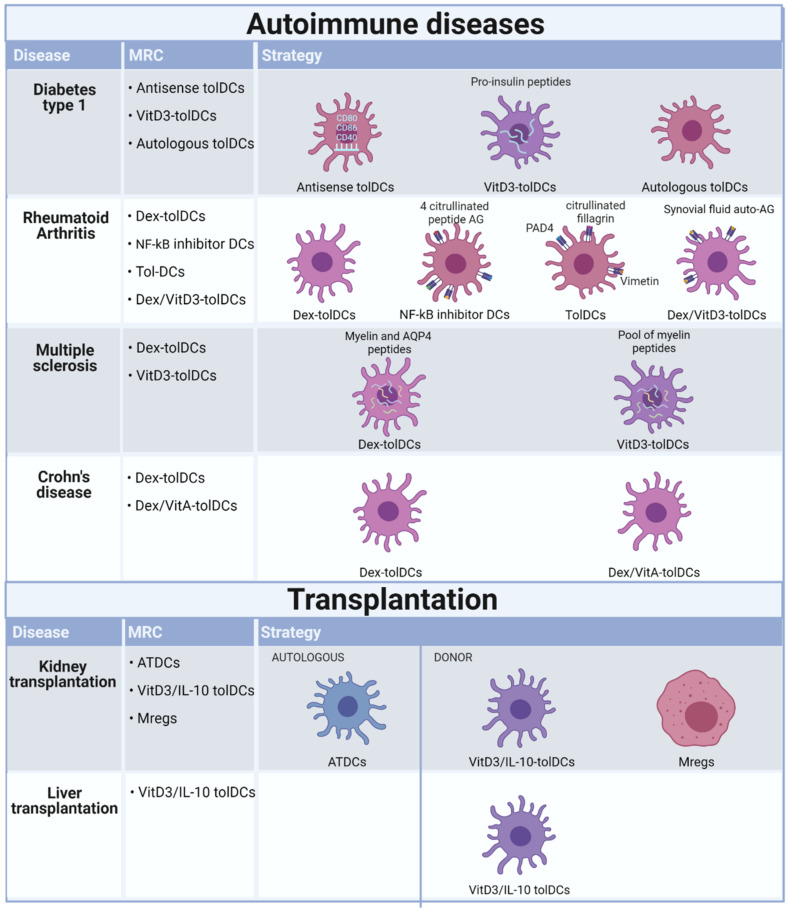
Tolerogenic dendritic cells (tolDCs) and regulatory macrophages (Mregs) studied in phase I/II clinical trials for several autoimmune diseases and kidney-/liver transplantation. Studies conducted for type I diabetes have used three different strategies, where the route of injection and frequency of injection differs per study. DIABETES MELLITUS TYPE I: monocyte-derived tolDCs were injected four times intradermally in a bi-weekly scheme [26]. The vitamin D3 (VitD3)-tolDCs loaded with proinsulin were injected two times intradermally with a 28-day interval. The autologous dendritic cells were injected intravenously in three monthly injections. RHEUMATOID ARTHRITIS: Dexamethasone (Dex)tolDCs were injected intra-arterially in a single infusion and dose escalation [27]. Nuclear factor kappa-light chain enhancer of activated B (NF-kB) inhibitor tolDCs, one dose of 2 progressive levels was injected intradermally [28].The route of injection of tolDCs pulsed with four different antigens are unknown but was injected five times in a two-dose regimen. Dex-VitD3-tolDCs loaded with synovial fluid were injected intralesionally in a dose-escalation via a single injection [29]. MULTIPLE SCLEROSIS: The Dex-tolDCs loaded with myelin peptides and aquaporin-4-derived peptides (AQP4) were injected in three injections, administered bi-weekly [30]. Two different clinical studies used the same subset of Dex-tolDCs loaded with a pool of myelin peptides but were injected either intranodally or intradermally. Both were dose escalation trials of six injections, four bi-weekly and two monthly injections [31]. CROHNS DISEASE: Dex-tolDCs were injected intralesionally (frequency unknown). Dex/Vitamin A (VitA)-tolDCs were injected intraperitoneally in a dose escalation of single injection versus three injections biweekly [33]. KIDNEY TRANSPLANTATION: Autologous tolDC (ATDC) was injected intravenously in a single injection one day before the transplantation. VitD3/interleukin (IL)-10 tolDCs were injected intravenously, single infusion one week before transplant. Mregs were injected intravenously 6/7 days before transplantation [32]. LIVER TRANSPLANTATION: VitD3/IL-10 tolDCs were injected intravenously in a single injection one week before immunosuppressive weaning. Created with BioRender.com.

**Table 1 ijms-22-07970-t001:** Completed and ongoing clinical trials using different tolDC subsets to treat autoimmune diseases.

ID	Status	Autoimmune Disease	Differentiation Protocol	Cohorts	Administration	Outcome	Ref.
**NCT00445913**	Completed	Type 1 diabetes	Tolerogenic monocyte-derived dendritic cells (Tol-MoDC) modified with anti-CD40, CD80, CD86 oligonucleotides (ODN)	Unmodified tol-monocyte derived DC (10 million cells) and ODN tol-MoDC (10 million cells)	Four intradermal injections, bi-weekly scheme	Increase in B220^+^CD11c B cells in blood and no adverse effects	[26]
**NTR5542**	Completed	Type 1 diabetes	Proinsuline-loaded vitamin D3 generated tolDCs	5, 10, 20 million cells	Two intradermal injections with a 28-day interval	unknown	
**NCT02354911**	Unknown	Type 1 diabetes	Tolerogenic monocyte-derived dendritic cells (Tol-MoDC) modified with anti-CD40, CD80, CD86 oligonucleotides (ODN)	10 million cells	Four intradermal injections, bi-weekly scheme		
**NCT03895996**	Recruiting	Type 1 diabetes	Autologous dendritic cell therapy	7–10 million cells	Three monthly intravenous injection		
**NCT03337165**	Completed	Rheumatoid arthritis	Dexamethasone generated tolDCs	1, 3, 5, 8, 10 million cells	Single intra-arterial infusion and dose escalation	Decrease in DAS28, HAQ improvement and no adverse effect	[27]
**Rheumavax**	Completed	Rheumatoid arthritis	Nuclear factor kappa-light chain enhancer of activated B (NF-kB) inhibitor tolDCs loaded with citrullinated peptides	low dose (1 million cells) and high dose (5 million cells)	One dose of 2 progressive levels was injected intradermally	Increase in Treg levels, decrease in T-cell response to vimentin 447–455, reduced serum level of proinflammatory cytokines/chemokines and no adverse effect	[28]
**CreaVax-RA**	Completed	Rheumatoid arthritis	DCs pulsed with PAD4, HNRNPA2B1, citrullinated filaggrin and vimentin antigens	0.5 or 1 million cells	Five injections times in a two-dose regimen	Decrease of IFNy-producing T cells and autoantibody levels and no adverse effect.	
**AutoDECRA**	Completed	Rheumatoid arthritis	Dexamethasone/vitaminD3 generated tolDCs loaded with autologous synovial fluid	1, 3, 10 million cells	Injection intralesional in a dose-escalation single injection	No biological effect in blood and no adverse effect	[29]
**NCT02283671**	Completed	Multiple sclerosis and Neuromyelitisoptica	Dexamethasone generated tolDCs loaded with myelin peptides or aquaporine-4- derived peptides	50, 100, 150, 300 million cells	Three injections administered bi-weekly	Decrease in frequency of CD8, NK, and CD14^+^ CD56^+^ cells and no adverse effect	[30]
**NCT02903537**	Recruiting	Multiple sclerosis	Vitamin D3 generated tolDCs loaded with a pool of myelin peptides	5, 10, 15 million cells	Intranodal injection in a dose escalation of six injections, four bi-weekly and two monthly injections		[31]
**NCT02618902**	Recruiting	Multiple sclerosis	Vitamin D3 generated tolDCs loaded with a pool of myelin peptides	5, 10, 15 million cells	Intradermal injection in a dose escalation of six injections, four bi-weekly and two monthly injections		[31]
**2007-003469-42**	Completed	Crohn’s disease	Dexamethasone/vitamin A generated tolDCs	2, 5, 10 million cells	Intralesional injection, frequency unkown	Patients withdrew due to worsening of symptoms and no adverse effect	[33]
**NCT02622763**	Terminated	Crohn’s disease	dexamethasone generated tolDCs	10, 100 million cells	Intraperitioneal injection in a dose escalation of single injections vs three injections biweekly		

**Table 2 ijms-22-07970-t002:** Completed and ongoing clinical trials using tolDC and Mregs in transplantation.

ID	Status	Transplantation	Differentiation Protocol	Cohorts	Administration	Outcome	Ref.
**NCT02252055**	Completed	Kidney	Granulocyte macrophage colony stimulating factor-stimulated tolDCs	1 million cells/kg	Intravenous injection in a single injection one day before transplantation	Administration of ATDCs are safe and feasible	[32]
**NCT02252055**	Active	Liver	Vitamin D3 and interleukin-10 generated tolDCs	2.5–10 million cells	Intravenous injection in a single infusion one week before transplantation		
**NCT03726307**	Recruiting	Kidney	Vitamin D3 and interleukin-10 generated tolDCs	0.5, 1.2, 2.5 million cells/kg	Intravenous injection in a single injection one week before transplantation		
**ONE study**	terminated	Kidney	Regulatory macrophages stimulated with macrophage colony stimulating factor and interferon-γ	2.5–7.5 million cells/kg	Intravenous injection in a single injection one week before transplantation	Administration of Mregs are safe and feasible	[32]

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
