# Peer review of "Regulatory Macrophages and Tolerogenic Dendritic Cells in Myeloid Regulatory Cell-Based Therapies"

_ijms, 2021, doi:10.3390/ijms22157970_

Round 1
Reviewer 1 Report
In the manuscript “Regulatory macrophages and tolerogenic dendritic cells in myeloid regulatory cell-based therapies”, the authors review the biology of tolerogenic myeloid cells describing the main factors driving their phenotype, mechanisms of T cell suppression, and results of preclinical and clinical studies in which they were employed. Investigation of myeloid cells’ immune regulatory properties represents an expanding field of research embracing cancer, autoimmunity and transplantation medicine. The review recapitulates results of therapeutic application of immune regulatory myeloid cells in autoimmune diseases and transplantation promoting further development of myeloid cell-based protocols. The authors should address the following concerns before achieving manuscript publication.
MAJOR CONCERNS:
- The authors should describe better the ancestry, phenotype and biological functions of regulatory macrophages: indeed this subset of macrophages can arise following different stimuli and show peculiar features that distinguish from both M1 and M2-like macrophages (David M. Mosser et al Nature Review Immunology 2008). Moreover, interferon γ has an acknowledged role in promoting the activation of macrophages towards a classical phenotype, thus the authors should reconcile the available literature on that topic.
- I strongly encourage the authors to recapitulate the results of clinical studies in a table describing the clinical trial phase, the adopted cell therapy, the results and the reference for each kind of disease.
Minor issues:
- Please revise the English language
- Please provide a description of acronyms in figure legends.
Author Response
REVIEWER 1:
In the manuscript “Regulatory macrophages and tolerogenic dendritic cells in myeloid regulatory cell-based therapies”, the authors review the biology of tolerogenic myeloid cells describing the main factors driving their phenotype, mechanisms of T cell suppression, and results of preclinical and clinical studies in which they were employed. Investigation of myeloid cells’ immune regulatory properties represents an expanding field of research embracing cancer, autoimmunity and transplantation medicine. The review recapitulates results of therapeutic application of immune regulatory myeloid cells in autoimmune diseases and transplantation promoting further development of myeloid cell-based protocols. The authors should address the following concerns before achieving manuscript publication.
MAJOR CONCERNS:
- The authors should describe better the ancestry, phenotype and biological functions of regulatory macrophages: indeed this subset of macrophages can arise following different stimuli and show peculiar features that distinguish from both M1 and M2-like macrophages (David M. Mosser et al Nature Review Immunology 2008). Moreover, interferon γ has an acknowledged role in promoting the activation of macrophages towards a classical phenotype, thus the authors should reconcile the available literature on that topic.
We thank the reviewer for this comment, We now added more details in macrophages in the Introduction part (pages 8-9 on the version with tracked changes), and completely rewrote the macrophage section to include their different origins, their classification including their phenotype and functions and their biological roles (pages 9-10+14). We mentioned the Mosser review as requested by the reviewer.
We also discussed the duality of IFN gamma and how this cytokine has been shown to promote tolerance and was consequently used to generate Mregs (pages 10 + 14).
- I strongly encourage the authors to recapitulate the results of clinical studies in a table describing the clinical trial phase, the adopted cell therapy, the results and the reference for each kind of disease.
We agree that tables describing the clinical trials allow a good view of the different trials (past and current trials). We added the tables in the review (pages 11-13) .
Minor issues:
- Please revise the English language
An English native speaker read the review and revised the English language.
- Please provide a description of acronyms in figure legends.
We apologize for this mistake, the acronyms are now explained in the legends of Figure 1 and Figure 2.

Reviewer 2 Report
I truly appreciate the authors’ efforts in writing this useful review about immunoregulatory functions and therapeutic applications of myeloid regulatory cells. The structure of the manuscript is logical, starting with introducing the importance of the problem, then presenting the mechanisms of different subtypes of tolerogenic dendritic cells and of regulatory macrophages, then their application in animal studies and human clinical trials, and finally their perspective for future research. Paragraph 2 – “Tolerogenic dendritic cells regulating peripheral tolerance” is very nicely approached and divided, with subparagraphs regarding the roles of vitamin D3, dexamethasone and rapamycin, as well as those assessing interleukin 10- dendritic cells, and autologous tolerogenic dendritic cells. It is a very comprehensive review, containing many details, written in an elegant manner. Despite of having so many details, data are easily to be followed. The authors included main required information and data are supported by many references, which are up-to-date. This field represents a domain which huge potential application in future practice. Some minor comments:
- Introduction:
- The sentence “The strong interest of DC is the ability to present antigens to T-cells and polarize an immune response” (lines 74-76) is somewhat redundant, given the extensive presentation of their functions above (lines 46-55). Maybe the authors would agree to delete it.
- Line 92: Please correct “Cell-based therapies is” to “Cell-based therapies are”.
- By the end of Introduction, please present the aim of the review in more details (as it appears from the abstract and the main text).
- Tolerogenic dendritic cells regulating peripheral tolerance: Line 158: Please correct “agent's” to “agents”
- Regulatory macrophages regulating peripheral tolerance: Lines 350-351: Please correct “M2 macrophages turned toward production ornithine” to “M2 macrophages turned toward production of ornithine”.
- I was thinking of two tables to summarize the findings of clinical trials, using myeloid regulatory cells in patients with autoimmune diseases (paragraph 4), respectively with transplantation (paragraph 5), but Figure 2 appears much more useful (as more explanations are given in the Figure 2 legend).
- “Future prosperities…”: I think “Future perspective(s)” would be much appropriate, as we do not know whether it would be prosperity or not. We just hope.
- Please insert places of Figures 1 and 2 in the main text.
- References:
- Reference 27 is not any more ahead of print. Please insert the update: 2021;105(4):832-841.
- Please correct and add to reference 88, the following data: 2000;30(7):1813-22. doi: 10.1002/1521-4141(200007)30:7<1813::AID-IMMU1813>3.0.CO;2-8.
- Please correct and add to reference 112, the following data: Curr Opin Organ Transplant. 2018;23(5):533-537. doi: 10.1097/MOT.0000000000000560.
Author Response
REVIEWER 2:
I truly appreciate the authors’ efforts in writing this useful review about immunoregulatory functions and therapeutic applications of myeloid regulatory cells. The structure of the manuscript is logical, starting with introducing the importance of the problem, then presenting the mechanisms of different subtypes of tolerogenic dendritic cells and of regulatory macrophages, then their application in animal studies and human clinical trials, and finally their perspective for future research. Paragraph 2 – “Tolerogenic dendritic cells regulating peripheral tolerance” is very nicely approached and divided, with subparagraphs regarding the roles of vitamin D3, dexamethasone and rapamycin, as well as those assessing interleukin 10- dendritic cells, and autologous tolerogenic dendritic cells. It is a very comprehensive review, containing many details, written in an elegant manner. Despite of having so many details, data are easily to be followed. The authors included main required information and data are supported by many references, which are up-to-date. This field represents a domain which huge potential application in future practice. Some minor comments:
- Introduction:
- The sentence “The strong interest of DC is the ability to present antigens to T-cells and polarize an immune response” (lines 74-76) is somewhat redundant, given the extensive presentation of their functions above (lines 46-55). Maybe the authors would agree to delete it.
We thank Reviewer 2 for this comment. We agree that this sentence was redundant and we delete it.
- Line 92: Please correct “Cell-based therapies is” to “Cell-based therapies are”.
We now modified the text accordingly.
- By the end of Introduction, please present the aim of the review in more details (as it appears from the abstract and the main text).
We agree that the aims of the review were not clearly defined at the end of the introduction and we now precise this part as follows (page 3 from the version with tracked changes, lines 100 to 112) “This review is focused on the therapeutic application of tolDCs and Mreg in autoimmune diseases and transplantation. The different subsets of tolDC (VitD3-tolDCs, Rapa-DC, DC10, Dex-tolDC and ATDC) and Mregs are described regarding their generation, their phenotype and their function. In the last sections, the completed and ongoing clinical trials using tolDC and Mregs are divided in transplantation and several autoimmune diseases, such as rheumatoid arthritis, type1 diabetes, multiple sclerosis, and Crohn's disease.”
- Tolerogenic dendritic cells regulating peripheral tolerance: Line 158: Please correct “agent's” to “agents”
We thank the reviewer for the correction and we changed it in the review.
- Regulatory macrophages regulating peripheral tolerance: Lines 350-351: Please correct “M2 macrophages turned toward production ornithine” to “M2 macrophages turned toward production of ornithine”.
We thank the reviewer for this remark. We rewrote the part on macrophages, due to modifications requested by Reviewer 1.
- I was thinking of two tables to summarize the findings of clinical trials, using myeloid regulatory cells in patients with autoimmune diseases (paragraph 4), respectively with transplantation (paragraph 5), but Figure 2 appears much more useful (as more explanations are given in the Figure 2 legend).
We agree that tables describing the clinical trials allow a good view of the different trials (past and current trials). We added the tables in the review (pages 11-13) .
- “Future prosperities…”: I think “Future perspective(s)” would be much appropriate, as we do not know whether it would be prosperity or not. We just hope.
We agree and adapted the title.
- Please insert places of Figures 1 and 2 in the main text.
Thank you for the reminder. We agree that it was missing in the text and added them.
- References:
- Reference 27 is not any more ahead of print. Please insert the update: 2021;105(4):832-841.
- Please correct and add to reference 88, the following data: 2000;30(7):1813-22. doi: 10.1002/1521-4141(200007)30:7<1813::AID-IMMU1813>3.0.CO;2-8.
- Please correct and add to reference 112, the following data: Curr Opin Organ Transplant. 2018;23(5):533-537. doi: 10.1097/MOT.0000000000000560.
We thank the reviewer for the corrections on the references. We modified these references.

Round 2
Reviewer 1 Report
The authors performed substantial modifications to the manuscript resulting in great improvements. All reviewer concerns have been promptly addressed. Only minor issues still need to be fixed before publication.
Minor issues:
- The manuscript still requires language revision: the text indeed has a remarkable amount of typing errors and sentences that need to be rephrased. Just to provide few examples, the following sentences need to be rephrased: lines 47-51, 414-419, 502-518, 522-524.
- I suggest modifying the sentences in lines 85-87 with the following one: “Lymphoid cell-based cancer immunotherapy strategies have been tested to restrict tumor progression by targeting tumor-associated antigens (PMID: 29152058, 26648934). Alternatively, approaches improving antigen-presenting cell (APC) abilities have been employed to re-educate host immune system to recognize tumor (PMID: 29752021, 15803149, 33907315)”.
- Please format the words “in vitro” and “in vivo” in italics and the symbol “+” in superscript when referred to a specific cell subset (e.g. “CD4+” in line 360).
Author Response
Reviewer 1:
The authors performed substantial modifications to the manuscript resulting in great improvements. All reviewer concerns have been promptly addressed. Only minor issues still need to be fixed before publication.
Minor issues:
- The manuscript still requires language revision: the text indeed has a remarkable amount of typing errors and sentences that need to be rephrased. Just to provide few examples, the following sentences need to be rephrased: lines 47-51, 414-419, 502-518, 522-524.
We thank the reviewer for this comment. We hired a professional that revised and reviewed the article on the English language.
- I suggest modifying the sentences in lines 85-87 with the following one: “Lymphoid cell-based cancer immunotherapy strategies have been tested to restrict tumor progression by targeting tumor-associated antigens (PMID: 29152058, 26648934). Alternatively, approaches improving antigen-presenting cell (APC) abilities have been employed to re-educate host immune system to recognize tumor (PMID: 29752021, 15803149, 33907315)”.
We agree with the modification of the sentences and changed them accordingly.
- Please format the words “in vitro” and “in vivo” in italics and the symbol “+” in superscript when referred to a specific cell subset (e.g. “CD4+” in line 360).
We agree with the suggested format and applied it to our text.
